# LEARNING TO DRIVE BY OBSERVING THE BEST AND SYNTHESIZING THE WORST

## ABSTRACT

Our goal is to train a policy for autonomous driving via imitation learning that is robust enough to drive a real vehicle. We find that standard behavior cloning is insufficient for handling complex driving scenarios, even when we leverage a perception system for preprocessing the input and a controller for executing the output on the car: 30 million examples are still not enough. We propose exposing the learner to synthesized data in the form of perturbations to the expert's driving, which creates interesting situations such as collisions and/or going off the road. Rather than purely imitating all data, we augment the imitation loss with additional losses that penalize undesirable events and encourage progress – the perturbations then provide an important signal for these losses and lead to robustness of the learned model. We show that the model can handle complex situations in simulation, and present ablation experiments that emphasize the importance of each of our proposed changes and show that the model is responding to the appropriate causal factors. Finally, we demonstrate the model driving a car in the real world.

## 1 INTRODUCTION

In order to drive a car, a driver needs to see and understand the various objects in the environment, predict their possible future behaviors and interactions, and then plan how to control the car in order to safely move closer to their desired destination while obeying the rules of the road. This is a difficult robotics challenge that humans solve well, making imitation learning a promising approach. Our work is about getting imitation learning to the level where it has a shot at driving a real vehicle; although the same insights may apply to other domains, these domains might have different constraints and opportunities, so we do not want to claim contributions there.

We built our system based on leveraging the training data (30 million real-world expert driving examples, corresponding to about 60 days of continual driving) as effectively as possible. There is a lot of excitement for end-to-end learning approaches to driving which typically focus on learning to directly predict raw control outputs such as steering or braking after consuming raw sensor input such as camera or lidar data. But to reduce sample complexity, we opt for mid-level input and output representations that take advantage of perception and control components. We use a perception system that processes raw sensor information and produces our input: a top-down representation of the environment and intended route, where objects such as vehicles are drawn as oriented 2D boxes along with a rendering of the road information and traffic light states. We present this mid-level input to a recurrent neural network (RNN), which then outputs a driving trajectory that is consumed by a controller which translates it to steering and acceleration. The further advantage of these mid-level representations is that the net can be trained on real or simulated data, and can be easily tested and validated in closed-loop simulations before running on a real car.

Our first finding is that even with 30 million examples, and even with mid-level input and output representations that remove the burden of perception and control, pure imitation learning is not sufficient. As an example, we found that this model would get stuck or collide with another vehicle parked on the side of a narrow street, when a nudging and passing behavior was viable. The key challenge is that we need to run the system closed-loop, where errors accumulate and induce a shift from the training distribution (Ross et al. (2011)). Scientifically, this result is valuable evidence about the limitations of pure imitation in the driving domain, especially in light of recent promising

results for high-capacity models (Laskey et al. (2017a)). But practically, we needed ways to address this challenge without exposing demonstrators to new states actively (Ross et al. (2011); Laskey et al. (2017b)) or performing reinforcement learning (Kuefler et al. (2017)).

We find that this challenge is surmountable if we augment the imitation loss with losses that discourage bad behavior and encourage progress, and, importantly, augment our data with *synthesized* perturbations in the driving trajectory. These expose the model to non-expert behavior such as collisions and off-road driving, and inform the added losses, teaching the model to avoid these behaviors. Note that the opportunity to synthesize this data comes from the mid-level input-output representations, as perturbations would be difficult to generate with either raw sensor input or direct controller outputs.

We evaluate our system, as well as the relative importance of both loss augmentation and data augmentation, first in simulation. We then show how our final model successfully drives a car in the real world and is able to negotiate situations involving other agents, turns, stop signs, and traffic lights. Finally, it is important to note that there are highly interactive situations such as merging which may require a significant degree of exploration within a reinforcement learning (RL) framework. This will demand simulating other (human) traffic participants, a rich area of ongoing research. Our contribution can be viewed as pushing the boundaries of what you can do with purely offline data and no RL.

## 2 RELATED WORK

Decades-old work on ALVINN (Pomerleau (1989)) showed how a shallow neural network could follow the road by directly consuming camera and laser range data. Learning to drive in an end-to-end manner has seen a resurgence in recent years. Recent work by Chen et al. (2015) demonstrated a convolutional net to estimate affordances such as distance to the preceding car that could be used to program a controller to control the car on the highway. Researchers at NVIDIA (Bojarski et al. (2016; 2017)) showed how to train an end-to-end deep convolutional neural network that steers a car by consuming camera input. Xu et al. (2017) trained a neural network for predicting discrete or continuous actions also based on camera inputs. Codevilla et al. (2017) also train a network using camera inputs and conditioned on high-level commands to output steering and acceleration. Kuefler et al. (2017) use Generative Adversarial Imitation Learning (GAIL) with simple affordance-style features as inputs to overcome cascading errors typically present in behavior cloned policies so that they are more robust to perturbations. Recent work from Hecker et al. (2018) learns a driving model using 360-degree camera inputs and desired route planner to predict steering and speed. The CARLA simulator (Dosovitskiy et al. (2017)) has enabled recent work such as Sauer et al. (2018), which estimates several affordances from sensor inputs to drive a car in a simulated urban environment. Using mid-level representations in a spirit similar to our own, Müller et al. (2018) train a system in simulation using CARLA by training a driving policy from a scene segmentation network to output high-level control, thereby enabling transfer learning to the real world using a different segmentation network trained on real data. Pan et al. (2017) also describes achieving transfer of an agent trained in simulation to the real world using a learned intermediate scene labeling representation. Reinforcement learning may also be used in a simulator to train drivers on difficult interactive tasks such as merging which require a lot of exploration, as shown in Shalev-Shwartz et al. (2016). A convolutional network operating on a space-time volume of bird's eye-view representations is also employed by Luo et al. (2018); Djuric et al. (2018); Lee et al. (2017) for tasks like 3D detection, tracking and motion forecasting. Finally, there exists a large volume of work on vehicle motion planning outside the machine learning context and Paden et al. (2016) present a notable survey.

## 3 MODEL ARCHITECTURE

### 3.1 INPUT OUTPUT REPRESENTATION

We begin by describing our top-down input representation that the network will process to output a drivable trajectory. At any time $t$, our agent (or vehicle) may be represented in a top-down coordinate system by $\mathbf{p}_t, \theta_t, s_t$, where $\mathbf{p}_t = (x_t, y_t)$ denotes the agent's location or pose, $\theta_t$ denotes the heading or orientation, and $s_t$ denotes the speed. The top-down coordinate system is picked such that our

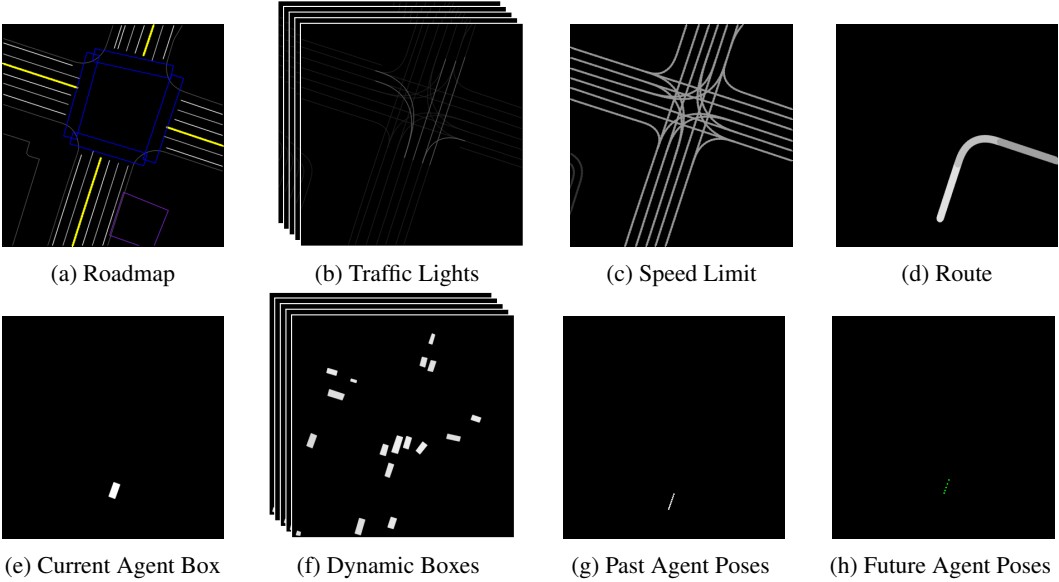

| (a) Roadmap | (b) Traffic Lights | (c) Speed Limit | (d) Route |
| (e) Current Agent Box | (f) Dynamic Boxes | (g) Past Agent Poses | (h) Future Agent Poses |

Figure 1: Driving model inputs (a-g) and output (h).

agent's pose $\mathbf{p}_0$ at the current time $t = 0$ is always at a fixed location $(u_0, v_0)$ within the image. The orientation of the coordinate system is randomly picked for each training example to be within an angular range of $\theta_0 \pm \Delta$, where $\theta_0$ denotes the heading or orientation of our agent at time $t = 0$.

As shown in Fig. 1, the input to our model consists of several images of size $W \times H$ pixels rendered into this top-down coordinate system. (a) Roadmap: a color (3-channel) image with a rendering of various map features such as lanes, stop signs, cross-walks, curbs, etc. (b) Traffic lights: a temporal sequence of grayscale images where each frame of the sequence represents the known state of the traffic lights at each past timestep. Within each frame, we color each lane center by a gray level with the brightest level for red lights, intermediate gray level for yellow lights, and a darker level for green or unknown lights[1]. (c) Speed limit: a single channel image with lane centers colored in proportion to their known speed limit. (d) Route: the intended route along which we wish to drive, generated by a router (think of a Google Maps-style route). (e) Current agent box: this shows our agent's full bounding box at the current timestep $t = 0$. (f) Dynamic objects in the environment: a temporal sequence of images showing all the potential dynamic objects (vehicles, cyclists, pedestrians) rendered as oriented boxes. (g) Past agent poses: the past poses of our agent are rendered into a single grayscale image as a trail of points.

We use a fixed-time sampling of $\delta t$ to sample any past or future temporal information, such as the traffic light state or dynamic object states in the above inputs. The traffic lights and dynamic objects are sampled over the past $T_{scene}$ seconds, while the past agent poses are sampled over a potentially longer interval of $T_{pose}$ seconds. This simple input representation, particularly the box representation of other dynamic objects, makes it easy to generate input data from simulation or create it from real-sensor logs using a standard perception system that detects and tracks objects. This enables testing and validation of models in closed-loop simulations before running them on a real car. This also allows the same model to be improved using simulated data to adequately explore rare situations such as collisions for which real-world data might be difficult to obtain. Using a top-down 2D view also means efficient convolutional inputs, and allows flexibility to represent metadata and spatial relationships in a human-readable format. Papers on testing frameworks such as Tian et al. (2018), Pei et al. (2017) show the brittleness of using raw sensor data (such as camera images or lidar point clouds) for learning to drive, and reinforce the approach of using an intermediate input representation.

---

[1]We employ an indexed representation for roadmap and traffic lights channels to reduce the number of input channels, and to allow extensibility of the input representation to express more roadmap features or more traffic light states without changing the model architecture.

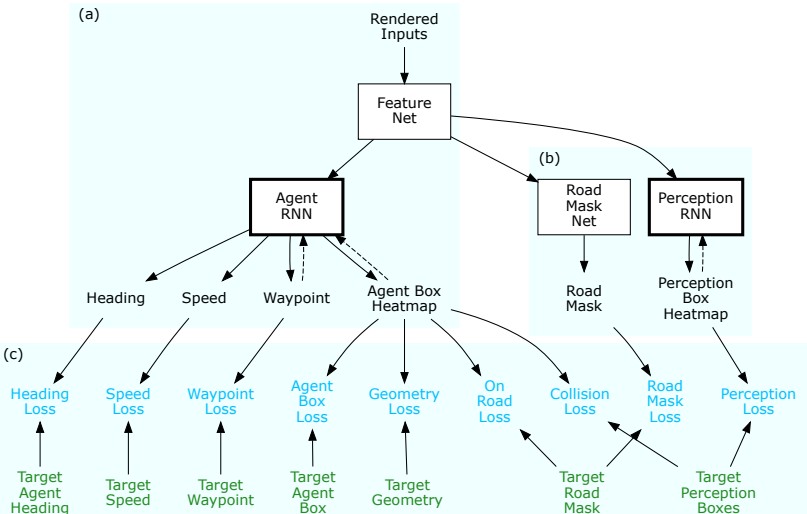

Figure 2: Training the driving model. (a) The core model with a FeatureNet and an AgentRNN, (b) Co-trained road mask prediction net and PerceptionRNN, and (c) Training losses are shown in blue, and the green labels depict the ground-truth data. The dashed arrows represent the recurrent feedback of predictions from one iteration to the next.

If $I$ denotes the set of all the inputs enumerated above, then the RNN recurrently predicts future poses of our agent conditioned on these input images $I$ as shown by the green dots in Fig. 1(h).

$$\mathbf{p}_{t+\delta t} = \text{RNN}(I, \mathbf{p}_t) \tag{1}$$

In Eq. (1), current pose $\mathbf{p}_0$ is a known part of the input, and then the RNN performs N iterations and outputs a future trajectory $\{\mathbf{p}_{\delta t}, \mathbf{p}_{2\delta t}, ..., \mathbf{p}_{N\delta t}\}$ along with other properties such as future speeds. This trajectory can be fed to a controls optimizer that computes detailed driving control (such as steering and braking commands) within the specific constraints imposed by the dynamics of the vehicle to be driven. Different types of vehicles may possibly utilize different control outputs to achieve the same driving trajectory, which argues against training a network to directly output low-level steering and acceleration control. Note, however, that having intermediate representations like ours does not preclude end-to-end optimization from sensors to controls.

### 3.2 MODEL DESIGN

Broadly, the driving model is composed of several parts as shown in Fig. 2. The main model shown in part (a) of the figure consists of a convolutional feature network (*FeatureNet*) that consumes the input data to create a digested contextual feature representation that is shared by the other networks. These features are consumed by a recurrent agent network (*AgentRNN*) that iteratively predicts successive points in the driving trajectory. Each point at time $t$ in the trajectory is characterized by its location $\mathbf{p}_t = (x_t, y_t)$, heading $\theta_t$ and speed $s_t$. The *AgentRNN* also predicts the bounding box of the vehicle as a spatial heatmap at each future timestep. In part (b) of the figure, we see that two other networks are co-trained using the same feature representation as an input. The Road Mask Network predicts the drivable areas of the field of view (on-road vs. off-road), while the recurrent perception network (*PerceptionRNN*) iteratively predicts a spatial heatmap for each timestep showing the future location of every other agent in the scene. We believe that doing well on these additional tasks using the same shared features as the main task improves generalization on the main task. Fig. 2(c) shows the various losses used in training the model, which we will discuss in detail below.

Fig. 3 illustrates the models in more detail. The rendered inputs shown in Fig. 1 are fed to a large-receptive field convolutional *FeatureNet* with skip connections, which outputs features $F$ that capture the environmental context and the intent. These features are fed to the *AgentRNN* which predicts the next point $\mathbf{p}_k$ on the driving trajectory, and the agent bounding box heatmap $B_k$, conditioned on the features $F$ from the FeatureNet, the iteration number $k \in \{1, \ldots, N\}$, the memory $M_{k-1}$

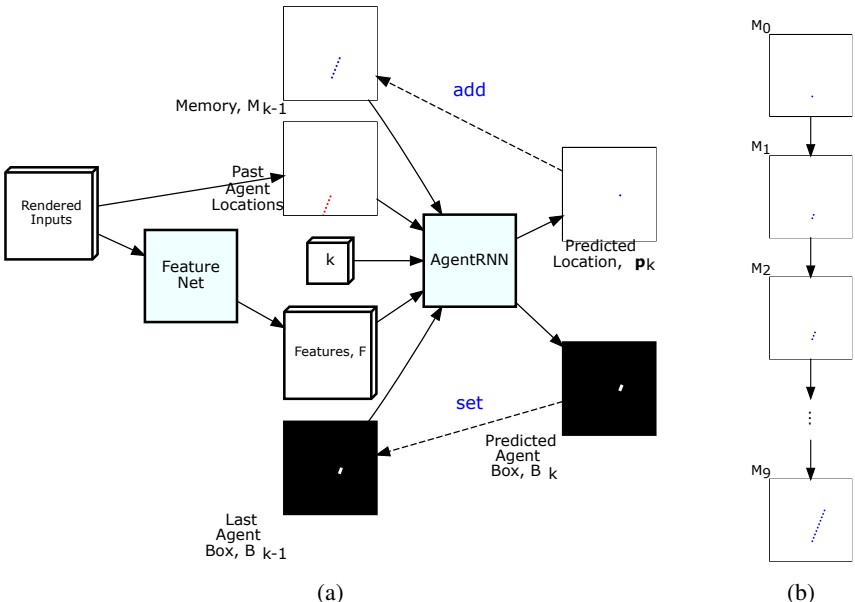

Figure 3: (a) Schematic of the AgentRNN. (b) Memory updates over multiple iterations.

of past predictions from the *AgentRNN*, and the agent bounding box heatmap $B_{k-1}$ predicted in the previous iteration.

$$\mathbf{p}_k, B_k = \text{AgentRNN}(k, F, M_{k-1}, B_{k-1}) \tag{2}$$

The memory $M_k$ is an additive memory consisting of a single channel image. At iteration $k$ of the *AgentRNN*, the memory is incremented by 1 at the location $\mathbf{p}_k$ predicted by the *AgentRNN*, and this memory is then fed to the next iteration. The *AgentRNN* outputs a heatmap image over the next pose of the agent, and we use the arg-max operation to obtain the coarse pose prediction $\mathbf{p}_k$ from this heatmap. The *AgentRNN* then employs a shallow convolutional meta-prediction network with a fully-connected layer that predicts a sub-pixel refinement of the pose $\delta\mathbf{p}_k$ and also estimates the heading $\theta_k$ and the speed $s_k$. Note that the *AgentRNN* is unrolled at training time for a fixed number of iterations, and the losses described below are summed together over the unrolled iterations. In the next section, we show how to train the model above to imitate the expert.

## 4  IMITATING THE EXPERT

### 4.1  IMITATION LOSSES

#### 4.1.1  AGENT POSITION, HEADING AND BOX PREDICTION

The *AgentRNN* produces three outputs at each iteration $k$: a probability distribution $P_k(x, y)$ over the spatial coordinates of the predicted waypoint obtained after a spatial *softmax*, a heatmap of the predicted agent box at that timestep $B_k(x, y)$ obtained after a per-pixel *sigmoid* activation that represents the probability that the agent occupies a particular pixel, and a regressed box heading output $\theta_k$. Given ground-truth data for the above predicted quantities, we can define the corresponding losses for each iteration as:

$$\mathcal{L}_p = \mathcal{H}(P_k, P_k^{gt}) \tag{3}$$

$$\mathcal{L}_B = \frac{1}{WH} \sum_x \sum_y \mathcal{H}(B_k(x, y), B_k^{gt}(x, y)) \tag{4}$$

$$\mathcal{L}_\theta = \left\| \theta_k - \theta_k^{gt} \right\|_1 \tag{5}$$

where the superscript *gt* denotes the corresponding ground-truth values, and $\mathcal{H}(a, b)$ is the cross-entropy function. Note that $P_k^{gt}$ is a binary image with only the pixel at the ground-truth target coordinate $\lfloor \mathbf{p}_k^{gt} \rfloor$ set to one.

### 4.1.2 AGENT META PREDICTION

The meta prediction network performs regression on the features to generate a sub-pixel refinement $\delta \mathbf{p}_k$ of the coarse waypoint prediction as well as a speed estimate $s_k$ at each iteration. We employ $L_1$ loss for both of these outputs:

$$\mathcal{L}_{p-subpixel} = \left\| \delta \mathbf{p}_k - \delta \mathbf{p}_k^{gt} \right\|_1 \tag{6}$$

$$\mathcal{L}_{speed} = \left\| s_k - s_k^{gt} \right\|_1 \tag{7}$$

where $\delta \mathbf{p}_k^{gt} = \mathbf{p}_k^{gt} - \lfloor \mathbf{p}_k^{gt} \rfloor$ is the fractional part of the ground-truth pose coordinates.

### 4.2 PAST MOTION DROPOUT

During training, the model is provided the past motion history as one of the inputs (Fig. 1(g)). Since the past motion history during training is from an expert demonstration, the net can learn to "cheat" by just extrapolating from the past rather than finding the underlying causes of the behavior. During closed-loop inference, this breaks down because the past history is from the net's own past predictions. For example, such a trained net may learn to only stop for a stop sign if it sees a deceleration in the past history, and will therefore never stop for a stop sign during closed-loop inference. To address this, we introduce a dropout on the past pose history, where for $50\%$ of the examples, we keep only the current position $(u_0, v_0)$ of the agent in the past agent poses channel of the input data. This forces the net to look at other cues in the environment to explain the future motion profile in the training example.

## 5 BEYOND PURE IMITATION

In this section, we go beyond vanilla cloning of the expert's demonstrations in order to teach the model to arrest drift and avoid bad behavior such as collisions and off-road driving by synthesizing variations of the expert's behavior.

### 5.1 SYNTHESIZING PERTURBATIONS

Running the model as a part of a closed-loop system over time can cause the input data to deviate from the training distribution. To prevent this, we train the model by adding some examples with realistic perturbations to the agent trajectories. The start and end of a trajectory are kept constant, while a perturbation is applied around the midpoint and smoothed across the other points. Quantitatively, we jitter the midpoint pose of the agent uniformly at random in the range $[-0.5, 0.5]$ meters in both axes, and perturb the heading by $[-\pi/3, \pi/3]$ radians. We then fit a smooth trajectory to the perturbed point and the original start and end points. Such training examples bring the car back to its original trajectory after a perturbation (see appendix for an example). We filter out some perturbed trajectories that are impractical by thresholding on maximum curvature. But we do allow the perturbed trajectories to collide with other agents or drive off-road, because the network can then experience and avoid such behaviors even though real examples of these cases are not present in the training data. In training, we give perturbed examples a weight of $1/10$ relative to the real examples, to avoid learning a propensity for perturbed driving.

### 5.2 BEYOND THE IMITATION LOSS

#### 5.2.1 COLLISION LOSS

Since our training data does not have any real collisions, the idea of avoiding collisions is implicit and will not generalize well. To alleviate this issue, we add a specialized loss that directly measures the overlap of the predicted agent box $B_k$ with the ground-truth boxes of all the scene objects at each timestep.

$$\mathcal{L}_{collision} = \frac{1}{WH} \sum_x \sum_y B_k(x, y) \cdot Obj_k^{gt}(x, y) \tag{8}$$

where $B_k$ is the likelihood map for the output agent box prediction, and $Obj_k^{gt}$ is a binary mask with ones at all pixels occupied by other dynamic objects (other vehicles, pedestrians, etc.) in the scene at timestep $k$. At any time during training, if the model makes a poor prediction that leads to a collision, the overlap loss would influence the gradients to correct the mistake. However, this loss would be effective only during the initial training rounds when the model hasn't learned to predict close to the ground-truth locations due to the absence of real collisions in the ground truth data. This issue is alleviated by the addition of trajectory perturbation data, where artificial collisions within those examples allow this loss to be effective throughout training without the need for online exploration like in reinforcement learning settings.

### 5.2.2 ON ROAD LOSS

Trajectory perturbations also create synthetic cases where the car veers off the road or climbs a curb or median because of the perturbation. To train the network to avoid hitting such hard road edges, we add a specialized loss that measures overlap of the predicted agent box $B_k$ in each timestep with a binary mask $Road^{gt}$ denoting the road and non-road regions within the field-of-view.

$$\mathcal{L}_{onroad} = \frac{1}{WH} \sum_x \sum_y B_k(x,y) . (1 - Road^{gt}(x,y)) \tag{9}$$

### 5.2.3 GEOMETRY LOSS

We would like to explicitly constrain the agent to follow the target geometry independent of the speed profile. We model this target geometry by fitting a smooth curve to the target waypoints and rendering this curve as a binary image in the top-down coordinate system. The thickness of this curve is set to be equal to the width of the agent. We express this loss similar to the collision loss by measuring the overlap of the predicted agent box with the binary target geometry image $Geom^{gt}$. Any portion of the box that does not overlap with the target geometry curve is added as a penalty to the loss function.

$$\mathcal{L}_{geom} = \frac{1}{WH} \sum_x \sum_y B_k(x,y) . (1 - Geom^{gt}(x,y)) \tag{10}$$

### 5.2.4 AUXILIARY LOSSES

Similar to our own agent's trajectory, the motion of other agents may also be predicted by a recurrent network. Correspondingly, we add a recurrent perception network *PerceptionRNN* that uses as input the shared features $F$ created by the *FeatureNet* and its own predictions $Obj_{k-1}$ from the previous iteration, and predicts a heatmap $Obj_k$ at each iteration. $Obj_k(x,y)$ denotes the probability that location $(x,y)$ is occupied by a dynamic object at time $k$. For iteration $k = 0$, the PerceptionRNN is fed the ground truth objects at the current time.

$$\mathcal{L}_{objects} = \frac{1}{WH} \sum_x \sum_y \mathcal{H}(Obj_k(x,y), Obj_k^{gt}(x,y)) \tag{11}$$

Co-training a *PerceptionRNN* to predict the future of other agents by sharing the same feature representation $F$ used by the *PerceptionRNN* is likely to induce the feature network to learn better features that are suited to both tasks.

We also co-train to predict a binary road/non-road mask by adding a small network of convolutional layers to the output of the feature net $F$. We add a cross-entropy loss to the predicted road mask output $Road(x,y)$ which compares it to the ground-truth road mask $Road^{gt}$.

$$\mathcal{L}_{road} = \frac{1}{WH} \sum_x \sum_y \mathcal{H}(Road(x,y), Road^{gt}(x,y)) \tag{12}$$

### 5.3 IMITATION DROPOUT

Overall, our losses may be grouped into two sub-groups: the imitation losses $\mathcal{L}_{imit} = \{\mathcal{L}_p, \mathcal{L}_B, \mathcal{L}_\theta, \mathcal{L}_{p-subpixel}, \mathcal{L}_{speed}\}$ and the environment losses $\mathcal{L}_{env} =$

| $T_{scene}$ | $T_{pose}$ | $\delta t$ | $N$ | $\Delta$ | $W$ | $H$ | $u_0$ | $v_0$ |
|---|---|---|---|---|---|---|---|---|
| 1.0 s | 8.0 s | 0.2s | 10 | $25°$ | 400 | 400 | 200 | 320 |

Table 1: Parameter values for the experiments in this paper.

$\{\mathcal{L}_{collision}, \mathcal{L}_{onroad}, \mathcal{L}_{geom}, \mathcal{L}_{objects}, \mathcal{L}_{road}\}$. The imitation losses cause the model to imitate the expert's demonstrations, while the environment losses discourage undesirable behavior such as collisions. To further increase the effectiveness of the environment losses, we experimented with randomly dropping out the imitation losses for a random subset of training examples. We refer to this as "imitation dropout". In the experiments, we show that imitation dropout yields a better driving model than simply under-weighting the imitation losses. During imitation dropout, the weight on the imitation losses $w_{imit}$ is randomly chosen to be either 0 or 1 with a certain probability for each training example. The overall loss is given by:

$$\mathcal{L} = w_{imit} \sum_{\ell \in \mathcal{L}_{imit}} \ell + w_{env} \sum_{\ell \in \mathcal{L}_{env}} \ell \tag{13}$$

# 6  EXPERIMENTS

## 6.1  MODELS

We train and test not only our final model, but a sequence of models that introduce the ingredients we describe one by one on top of behavior cloning. We start with $\mathcal{M}_0$, which does behavior cloning with past motion dropout to prevent using the history to cheat. $\mathcal{M}_1$ adds perturbations without modifying the losses. $\mathcal{M}_2$ further adds our environment losses $\mathcal{L}_{env}$ in Section 5.2. $\mathcal{M}_3$ and $\mathcal{M}_4$ address the fact that we do not want to imitate bad behavior – $\mathcal{M}_3$ is a baseline approach, where we simply decrease the weight on the imitation loss, while $\mathcal{M}_4$ uses our imitation dropout approach with a dropout probability of 0.5. Table 2 lists the configuration for each of these models.

## 6.2  DATA

The training data to train our model was obtained by randomly sampling segments of real-world expert driving and removing segments where the car was stationary for long periods of time. Since our input field of view is $80m \times 80m$, we also removed any segments of highway driving for the experiments presented here, because highway driving would need a larger field of view. Our dataset contains approximately 26 million examples which amount to about 60 days of continuous driving. As discussed in Section 3, the vertical-axis of the top-down coordinate system for each training example is randomly oriented within a range of $\Delta = \pm 25°$ of our agent's current heading, in order to avoid a bias for driving along the vertical axis. The rendering orientation is set to the agent heading ($\Delta = 0$) during inference. Data about the prior map of the environment (roadmap) and the speed-limits along the lanes is collected apriori. For the dynamic scene entities like objects and traffic-lights, we employ a separate perception system based on laser and camera data similar to existing works in the literature (Fairfield & Urmson (2011); Levinson et al. (2011)). Table 1 lists the parameter values used for all the experiments in this paper.

## 6.3  EVALUATION

To evaluate our learned model on a specific scenario, we replay the segment through the simulation until a buffer period of $\max(T_{pose}, T_{scene})$ has passed. This allows us to generate the first rendered snapshot of the model input using all the replayed messages until now. The model is evaluated on this input, and the fitted controls are passed to the vehicle simulator that emulates the dynamics of the vehicle thus moving the simulated agent to its next pose. At this point, the simulated pose might be different from the logged pose, but our input representation allows us to correctly render the new input for the model relative to the new pose. This process is repeated until the end of the segment, and we evaluate scenario specific metrics like stopping for a stop-sign, collision with another vehicle etc. during the simulation. Since the model is being used to drive the agent forward, this is a *closed-loop* evaluation setup.

| Model | Description | $w_{imit}$ | $w_{env}$ |
|---|---|---|---|
| $\mathcal{M}_0$ | Imitation with Past Dropout | 1.0 | 0.0 |
| $\mathcal{M}_1$ | $\mathcal{M}_0$ + Traj Perturbation | 1.0 | 0.0 |
| $\mathcal{M}_2$ | $\mathcal{M}_1$ + Environment Losses | 1.0 | 1.0 |
| $\mathcal{M}_3$ | $\mathcal{M}_2$ with less imitation | 0.5 | 1.0 |
| $\mathcal{M}_4$ | $\mathcal{M}_2$ with Imitation Dropout | *Dropout probability = 0.5 (see Section 5.3)*. | |

Table 2: Model configuration for the model ablation tests.

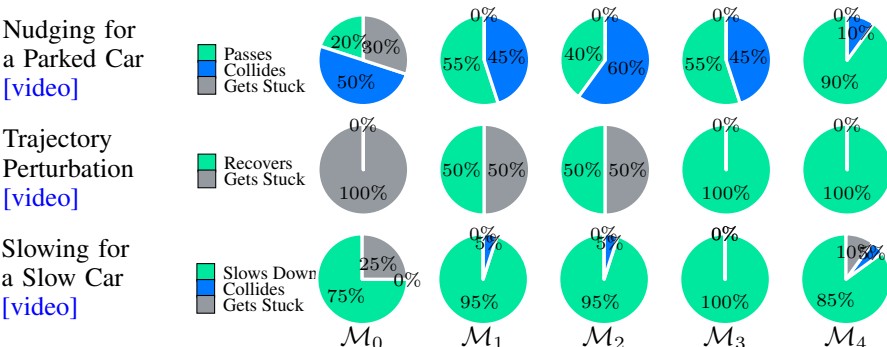

Figure 4: Model ablation test results on three scenario types.

### 6.3.1 MODEL ABLATION TESTS

Here, we present results from experiments using the various models in the closed-loop simulation setup. We first evaluated all the models on simple situations such as stopping for stop-signs and red traffic lights, and lane following along straight and curved roads by creating 20 scenarios for each situation, and found that *all the models worked well in these simple cases*. Therefore, we will focus below on specific complex situations that highlight the differences between these models.

**Nudging around a parked car.** To set up this scenario, we place the agent at an arbitrary distance from a stop-sign on an undivided two-way street and then place a parked vehicle on the right shoulder between the the agent and the stop-sign. We pick 4 separate locations with both straight and curved roads then vary the starting speed of the agent between 5 different values to create a total of 20 scenarios. We then observe if the agent would stop and get stuck behind, collide with the parked car, or correctly pass around the parked car, and report the aggregate performance in Fig. 4(row 1). We find that other than $\mathcal{M}_4$, all other models cause the agent to collide with the parked vehicle about half the time. The baseline $\mathcal{M}_0$ model can also get stuck behind the parked vehicle in some of the scenarios. The model $\mathcal{M}_4$ nudges around the parked vehicle and then brings the agent back to the lane center. This can be attributed to the model's ability to learn to avoid collisions and nudge around objects because of training with the collision loss the trajectory perturbation. Comparing model $\mathcal{M}_3$ and $\mathcal{M}_4$, it is apparent that "imitation dropout" was more effective at learning the right behavior than only re-weighting the imitation losses.

**Recovering from a trajectory perturbation.** To set up this scenario, we place the agent approaching a curved road and vary the starting position and the starting speed of the agent to generate a total of 20 scenario variations. Each variation puts the agent at a different amount of offset from the lane center with a different heading error relative to the lane. We then measure how well the various models are at recovering from the lane departure. Fig. 4(row 2) presents the results aggregated across these scenarios and shows the contrast between the baseline model $\mathcal{M}_0$ which is not able to recover in any of the situations and the models $\mathcal{M}_3$ and $\mathcal{M}_4$ which handle all deviations well. All models trained with the perturbation data are able to handle 50% of the scenarios which have a lower starting speed. At a higher starting speed, we believe that $\mathcal{M}_3$ and $\mathcal{M}_4$ do better than $\mathcal{M}_1$ and $\mathcal{M}_2$ because they place a higher emphasis on the imagination losses.

**Slowing down for a slow car.** To set up this scenario, we place the agent on a straight road at varying initial speeds and place another car ahead with a varying but slower constant speed, generating a total of 20 scenario variations, to evaluate the ability to slow for and then follow the car ahead. From Fig. 4(row 3), we see that some models slow down to zero speed and get stuck. For the

variation with the largest relative speed, there isn't enough time for most models to stop the agent in time, thus leading to a collision. For these cases, model $\mathcal{M}_3$ which uses imitation loss re-weighting works better than the model $\mathcal{M}_4$ which uses imitation dropout. $\mathcal{M}_4$ has trouble in two situations due to being over aggressive in trying to maneuver around the slow car and then grazes the left edge.

### 6.3.2 INPUT ABLATION TESTS

With input ablation tests, we want to test the final $\mathcal{M}_4$ model's ability to identify the correct causal factors behind specific behaviors, by testing the model's behavior in the presence or absence of the correct causal factor while holding other conditions constant. In simulation, we have evaluated our model on 20 scenarios with and without stop-signs rendered, and 20 scenarios with and without other vehicles in the scene rendered. The model exhibits the correct behavior in all scenarios, thus confirming that it has learned to respond to the correct features for a stop-sign and a stopped vehicle.

### 6.3.3 OPEN LOOP EVALUATION

In an open-loop evaluation, we take test examples of expert driving data and for each example, compute the $L_2$ distance error between the predicted and ground-truth waypoints. Unlike the closed-loop setting, the predictions are not used to drive the agent forward and thus the network never sees its own predictions as input. Fig. 5 shows the $L_2$ distance metric in this open-loop evaluation setting for models $\mathcal{M}_0$ and $\mathcal{M}_4$ on a test set of 10,000 examples. These results show that model $\mathcal{M}_0$ makes fewer errors than the full model $\mathcal{M}_4$, but we know from closed-loop testing that $\mathcal{M}_4$ is a far better driver than $\mathcal{M}_0$. This shows how open-loop evaluations can be misleading, and closed-loop evaluations are critical while assessing the real performance of such driving models.

Figure 5: Prediction Error for models $\mathcal{M}_0$ and $\mathcal{M}_4$ on unperturbed evaluation data.

### 6.3.4 REAL WORLD DRIVING

We have also evaluated this model on our self-driving car by replacing the existing planner module with the learned model $\mathcal{M}_4$ and have replicated the driving behaviors observed in simulation. The videos of several of these runs are available here.

### 6.4 FAILURE MODES

At our ground resolution of 20 cm/pixel, the agent currently sees 64 m in front and 40 m on the sides and this limits the model's ability to perform merges on T-junctions and turns from a high-speed road. Specific situations like U-turns and cul-de-sacs are also not currently handled, and will require sampling enough training data. The model occasionally gets stuck in some low speed nudging situations. It sometimes outputs turn geometries that make the specific turn infeasible (e.g. large turning radius). We also see some cases where the model gets over aggressive in novel and rare situations for example by trying to pass a slow moving vehicle. We believe that adequate simulated exploration may be needed for highly interactive or rare situations.

## 7 DISCUSSION

In this paper, we presented our experience with what it took to get imitation learning to perform well in real-world driving. We found that key to its success is synthesizing interesting situations around the expert's behavior and augmenting appropriate losses that discourage undesirable behavior. This constrained exploration is what allowed us to avoid collisions and off-road driving even though such examples were not explicitly present in the expert's demonstrations. To support it, and to best leverage the expert data, we used middle-level input and output representations which allow easy mixing of real and simulated data and alleviate the burdens of learning perception and control. With these ingredients, we got a model good enough to drive a real car. That said, there is room for improvements: comparing to end-to-end approaches, or investigating alternatives to imitation dropout are among them. But most importantly, we believe that augmenting the expert demonstrations with a thorough exploration of rare and difficult scenarios in simulation, perhaps within a reinforcement learning framework, will be the key to improving the performance of these models especially for highly interactive scenarios.

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

| Rendering | FeatureNet | AgentRNN (N=10) | PerceptionRNN (N=10) | **Overall** |
|---|---|---|---|---|
| 8 ms | 6.5 ms | 145 ms | 35 ms | **160 ms** |

Table 3: Run-time performance on NVIDIA Tesla P100 GPU.

## APPENDIX

### A    SYSTEM ARCHITECTURE

Fig. 6 shows a system level overview of how the neural net is used within the self-driving system. At each time, the updated state of our agent and the environment is obtained via a perception system that processes sensory output from the real-world or from a simulation environment as the case may be. The intended route is obtained from the router, and is updated dynamically conditioned on whether our agent was able to execute past intents or not. The environment information is rendered into the input images described in Fig. 1 and given to the RNN which then outputs a future trajectory. This is fed to a controls optimizer that outputs the low-level control signals that drive the vehicle (in the real world or in simulation).

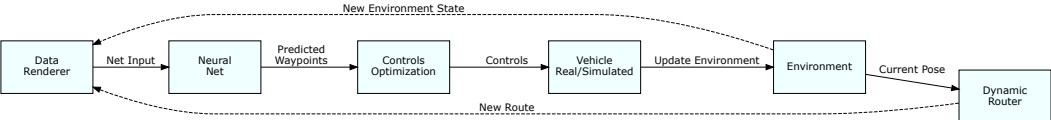

Figure 6: Software architecture for the end-to-end driving pipeline.

### B    PERFORMANCE

The model runs on a NVidia Tesla P100 GPU in 160ms with the detailed breakdown in Table 3.

### C    VISUAL EXAMPLES

Fig. 7 shows an example of perturbing the current agent location (red point) away from the lane center and the fitted trajectory correctly bringing it back to the original target location along the lane center. Fig. 8 shows the various predictions and losses for a single example processed through the model.

### D    OPEN LOOP EVALUATION ON PERTURBATION DATA

We also compare the performance of models $\mathcal{M}_0$ and $\mathcal{M}_1$ on our perturbed evaluation data w.r.t the $L_2$ distance metric in Fig. 9. Note that the model trained without including perturbed data ($\mathcal{M}_0$) has larger errors due to its inability to bring the agent back from the perturbation onto its original trajectory. Fig. 10 shows examples of the trajectories predicted by these models on a few representative examples showcasing that the perturbed data is critical to avoiding the veering-off tendency of the model trained without such data.

### E    SAMPLING SPEED PROFILES

The waypoint prediction from the model at timestep $k$ is represented by the probability distribution $P_k(x, y)$ over the spatial domain in the top-down coordinate system. In this paper, we pick the mode of this distribution $\mathbf{p}_k$ to update the memory of the $AgentRNN$. More generally, we can also sample from this distribution to allow us to predict trajectories with different speed profiles. Fig. 11 illustrates the predictions $P_1(x, y)$ and $P_5(x, y)$ at the first and the fifth iterations respectively, for a training example where the past motion history has been dropped out. Correspondingly, $P_1(x, y)$ has a high uncertainty along the longitudinal position and allows us to pick from a range of speed samples. Once we pick a specific sample, the ensuing waypoints get constrained in their ability to pick different speeds and this shows as a centered distribution at the $P_5(x, y)$.

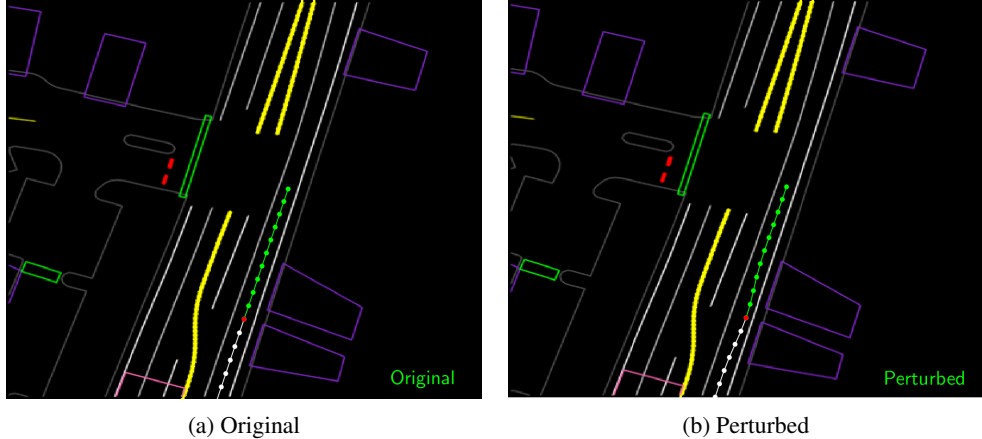

|  |  |
|---|---|
| (a) Original | (b) Perturbed |

Figure 7: Trajectory Perturbation. (a) An original logged training example where the agent is driving along the center of the lane. (b) The perturbed example created by perturbing the current agent location (red point) in the original example away from the lane center and then fitting a new smooth trajectory that brings the agent back to the original target location along the lane center.

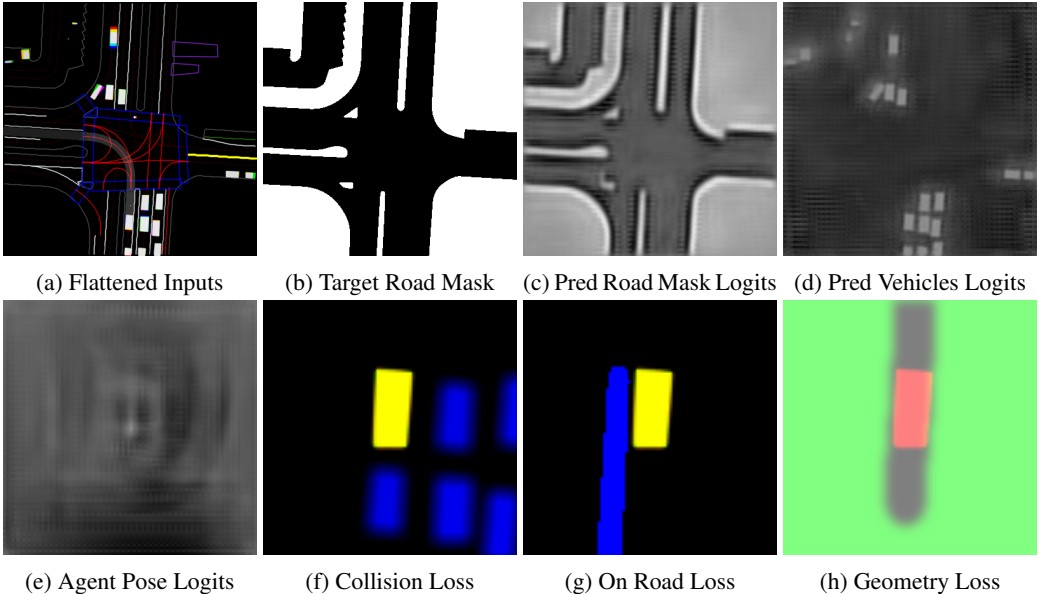

| (a) Flattened Inputs | (b) Target Road Mask | (c) Pred Road Mask Logits | (d) Pred Vehicles Logits |
|---|---|---|---|
| (e) Agent Pose Logits | (f) Collision Loss | (g) On Road Loss | (h) Geometry Loss |

Figure 8: Visualization of predictions and loss functions on an example input. The top row is at the input resolution, while the bottom row shows a zoomed-in view around the current agent location.

The use of a probability distribution over the next waypoint also presents the interesting possibility of constraining the model predictions at inference time to respect hard constraints. For example, such constrained sampling may provide a way to ensure that any trajectories we generate strictly obey legal restrictions such as speed limits. One could also constrain sampling of trajectories to a designated region, such as a region around a given reference trajectory.

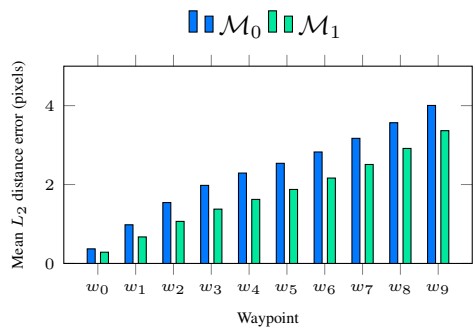

Figure 9: Prediction Error for models $\mathcal{M}_0$ and $\mathcal{M}_1$ on perturbed evaluation data.

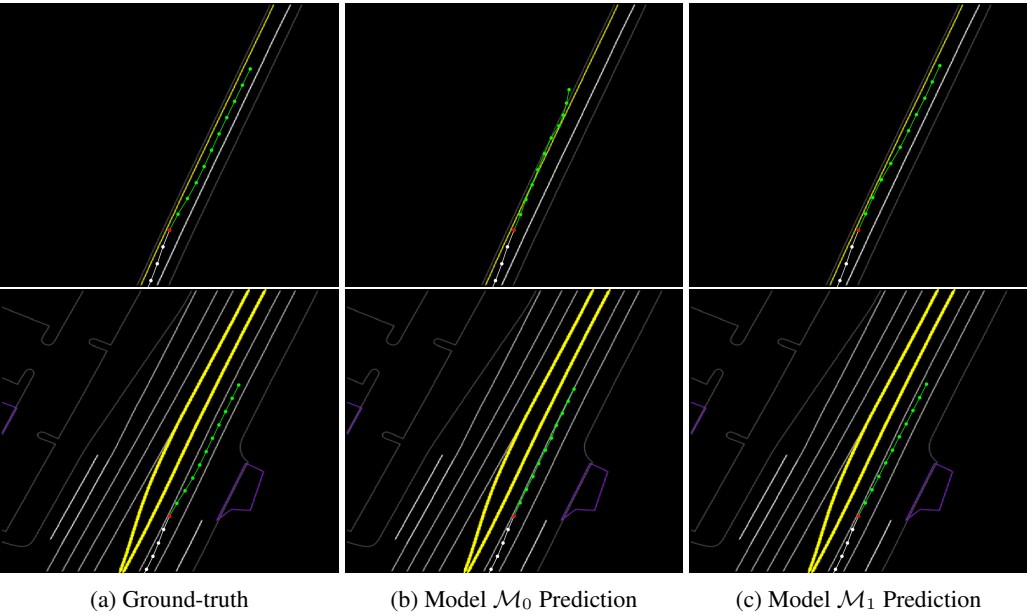

(a) Ground-truth      (b) Model $\mathcal{M}_0$ Prediction      (c) Model $\mathcal{M}_1$ Prediction

Figure 10: Comparison of ground-truth trajectory in (a) with the predicted trajectories from models $\mathcal{M}_0$ and $\mathcal{M}_1$ in (b) and (c) respectively on two perturbed examples. The red point is the reference pose $(u_0, v_0)$, white points are the past poses and green points are the future poses.

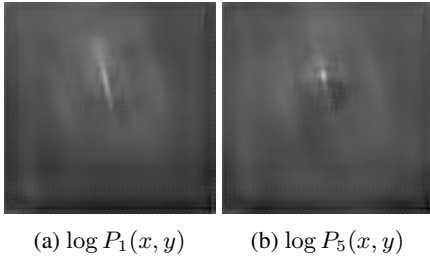

(a) $\log P_1(x, y)$      (b) $\log P_5(x, y)$

Figure 11: Sampling speed profiles. The probability distribution $P_1(x, y)$ predicted by the model at timestep $k = 1$ allows us to sample different speed profiles conditioned on which the later distribution $P_5(x, y)$ gets more constrained.

