# OpenReview forum: "Learning to Drive by Observing the Best and Synthesizing the Worst"
_ICLR.cc/2019/Conference_

### Official Review · AnonReviewer1 · 2018-11-02
**A reasonable approach for self-driving vehicle control.**

**Rating:** 5
**Confidence:** 4

**Review:**

Summary.
The paper proposes a vehicle’s trajectory planner that iteratively predict next-step (longitudinal and latitudinal) position of an ego-vehicle. Instead of using a raw image, a set of handcrafted features (i.e., the status of traffic lights, route, roadmap, etc) are mapped onto a fixed-size of bird-eye view map, which is then fed into the recurrent neural network. Additional regularizing loss terms are explored for the robustness of the model. The effectiveness of the method is demonstrated in simulation and real-world experiment.

Strengths.
- Impressive demonstrations in simulation and real-world experiments.
- The paper is generally well-written and easy to follow.

vs. Existing motion planning approaches.
There exists a large volume of papers on vehicle motion planning, which has largely been explored for controlling self-driving vehicles. Some of them successfully demonstrated their effectiveness for navigating a vehicle in typical driving scenarios, including “slowing down for a slow car”.
A notable survey may include:

[1] Paden et al., “A survey of motion planning and control techniques for self-driving urban vehicles,” IEEE Transactions on intelligent vehicles, 2016.

However, the paper provides neither any works of literature on existing motion planners nor any types of comparison with them. This makes hard to judge the proposed learning-based motion planner outperforms others including conventional optimization-based methods.

Missing data collection details.
This work depends hugely on its own human-designated oracle-like map, which provides driving-related features, such as lane, the status of traffic lights, speed limits, desired route, dynamic objects, etc. Generating this map would not be a trivial task, but details are missing on (1) how this data collected and (2) how this data can be collected during the testing time (especially for dynamic objects/traffic light status). Section 6.2 should be explained more in detail.

A weak novelty of using intermediate-level input/output representation.
There exist similar approaches that utilized similar representations to determine a vehicle’s behaviour, examples may include:

[1] Lee et al., “Convolution Neural Network-based Lane Change Intention Prediction of Surrounding Vehicles for ACC,” IEEE ITSC 2017.
[2] We et al., “Modeling trajectories with recurrent neural networks,” IJCAI, 2017.

Missing evaluation details.
In Section 6.2, (though not mentioned) it seems that a training dataset is collected from 60-days of real-world driving (given the context). But, in the testing phase, it seems that the authors used a simulator to evaluate different driving scenarios with various initial condition (i.e., speed, heading angle, position, etc). Can authors clarify details of the evaluation environment?

Minor concerns.
A paragraph of contribution summary (in Introduction section) will help.
Typos (e.g., Section 2 line 17: ‘off of’)

---

> ### Author Response · Authors · 2018-11-10
> **Author Response**
>
> Thanks for reviewing the paper and for your valuable feedback.
>
> “Existing motion planning approaches”: We have included this reference in the revised draft. Motion planning is really useful in cases where the cost function, constraints, and dynamics are clear; none of that is true for driving -- writing down the true cost we want optimized is hard and contextual, and the dynamics are hard especially because they involve other people in the environment and what they will do; we thus think it's useful to see how far imitation learning can be pushed as an alternative. We would also like to emphasize that this is not only useful for driving the car, but also can be integrated within a motion planner as a model of how other people will act.
>
> “Data collection details”: Data about the prior map of the environment (roadmap) and the speed-limits along the lanes is collected apriori. For the dynamic scene entities like objects and traffic-lights, we employ a separate perception system based on laser and camera data similar to existing works in the literature [1,2]. We have clarified this in the revised version.
>
> [1] Fairfield, Nathaniel, and Chris Urmson. "Traffic light mapping and detection." Robotics and Automation (ICRA), 2011 IEEE International Conference on. IEEE, 2011.
> [2] Yang, Bin, Ming Liang, and Raquel Urtasun. "HDNET: Exploiting HD Maps for 3D Object Detection." Conference on Robot Learning. 2018.
>
> “Other references”: Our paper includes recent references to works on autonomous driving using the mid-level input/output representations. In the revised version, we have also included the suggested references which specifically target use-cases like predicting other agents’ intent.
>
> “Evaluation details”: The training data is collected from real-world driving. We perform testing on these kinds of data:
>
> Simulated Data: For this evaluation, we create specific scenarios as described in section 6.3.1 within our simulator to allow us to test specific conditions without introducing the complexities of real world all at the same time and the quantitative results in Fig. 4 point to these results [ https://sites.google.com/view/learn-to-drive#h.p_XLYMjRiONt1e ].
>
> Logged Data: For this evaluation, we take logs from our real-driving test data (separate from our training data), and use our trained network to drive the car using the vehicle simulator keeping everything else the same i.e. the dynamic objects, traffic-light states etc. are all kept the same as in the logs. These drives are shown in the supplemental website [ https://sites.google.com/view/learn-to-drive#h.p_cxFQRIZYOQ7o ].
>
> Logged Ablation Data: This is the same as above, except that we modify some of the rendered inputs like removing the stop-signs or other dynamic objects to generate the input ablation results in section 6.3.2 and the videos on the supplemental site [ https://sites.google.com/view/learn-to-drive#h.p_WjtxfxJsNmJT ].
>
> Real Drive: This is where we let the network drive a real car [ https://sites.google.com/view/learn-to-drive#h.p_zId3Ux6DONGv ].
>
> “Contribution Summary”: We have updated the introduction to clarify the contributions.

---

### Official Review · AnonReviewer3 · 2018-11-05
**Interesting examples, somewhat weaker evaluation**

**Rating:** 6
**Confidence:** 4

**Review:**

The authors present a very interesting work on predicting future motion of a self-driving vehicle given image inputs that represent its surrounding and history. The authors use RNNs for this task, and data augmentation to make their model more robust. They also present a number of very interesting videos showcasing the performance.
- Several key aspects of the work are not well explained. E.g., what are pixel sizes, time resolution, where is the vehicle positioned within the image? All this is missing.
- Traffic lights are represented as "a sequence of grayscale images", how exactly, one for each state? Or some other way.
- How were videos generated, how were various channels collapsed?
- Dashed arrows not explained in Fig 2.
- "a small regression tower", this needs to be elaborated. As well as other mentions of "towers".
- In (3), is the sum over all pixels missing?
- Section 4.1.2 is not clear, this needs to be expanded. It is not well explained how exactly these losses are computed and used.
- For past-motion dropout, then you simply give blank input?
- Figure 6 is referenced in the regular text although it is located in the appendix.
- Orienting vertical axis with delta of +-25deg (as explained in Section 6.2) is not observed in the given videos, seems that there is no delta there. Is that done only during training?
- What is the exact difference between open- and closed-loop experiments? Given that a number of other key aspects are missing, I am not sure I fully understand a difference here as well.
- One of major issues in the evaluation section is that other baselines are missing (especially in the context of Fig 5). Even the more obvious ones would help a lot with understanding the performances, such as vehicle continuing to do what it was doing, or baseline predicting the route). This is a major flaw of the paper.
- Some recent related work missing, see [1], [2], and related work.
[1] Fast and Furious: Real Time End-to-End 3D Detection, Tracking and Motion Forecasting With a Single Convolutional Net, Luo, Wenjie, Bin Yang, and Raquel Urtasun, Proceedings of the IEEE Conference on Computer Vision and Pattern Recognition. 2018.
[2] Short-Term Motion Prediction of Traffic Actors for Autonomous Driving using Deep Convolutional Networks, Djuric, N., Radosavljevic, V., Cui, H., Nguyen, T., Chou, F.-C., Lin, T.-H., Schneider, J., arXiv preprint:1808.05819, 2018.

---

> ### Author Response · Authors · 2018-11-10
> **Author Response (Part 1 of 2)**
>
> Thank you for reviewing the paper and for your valuable feedback. We have uploaded a revised version that clarifies the technical details but to answer your questions more directly:
>
> “Pixel sizes, time resolution, vehicle position etc.”: As suggested in the author guidelines, we listed these details in Table 3 in the Appendix as we considered them important for reproducibility but not for the core understanding of the paper. We have moved this table into the main text in the revised version. We will appreciate your feedback on any other key details that we might have missed.
>
> “Traffic lights”: The sequence dimension represents past timesteps like the other channels. Each frame in the sequence is a gray-scale image with a specific intensity representing a specific traffic light state for each lane. For example, we use an intensity of 96 to represent a green signal and 224 to represent a red signal. We have revised the description in the new paper version.
>
> “Videos”: Each frame of the video represents the current state of the environment as follows: video_t = max(roadmap_t, route_t, current_agent_box_t, past_agent_poses_t, predicted_future_poses_{1,10}, traffic_lights_t, dynamic_boxes_past), with specific mapping of specific channels to individual colors in the output for ease of viewing. To aid visualization of the movement of dynamic objects, we combine the sequence of dynamic_boxes_{t-1,...,t} using a weighted sum to generate the dynamic_boxes_past channel above.
>
> “Dashed arrows”: The dashed arrows represent the recurrent feedback of predictions from iteration k as inputs into iteration k+1. We have clarified this in the paper now.
>
> “Regression tower”: We use a shallow convolutional network with a fully-connected layer at the end for this. We use “towers” to imply any building block consisting of a sequence of basic convolutional, space-to-depth, depth-to-space, fully-connected and activation layers. We have clarified this in the paper.
>
> “Eqn 3”: The logits P_k represent a probability distribution and the cross-entropy function H thus computes a single cross-entropy value for the input P_k and P_k^gt without the need for a summation over pixels.
>
> “Section 4.1.2”: We have added further details to the revised version. The prediction P_k is a probability distribution over the next predicted waypoint and we pick and arg-max over this distribution to sample an integer coordinate. To refine this to sub-pixel resolution, the agent-meta prediction network produces subpixel values \delta{p_k} = (\delta_x, \delta_y) which are then compared with the ground-truth values using L1 norm as in equation 6. Similarly, it produces a speed value s_k which is compared with the target value using L1 norm as well. Theses losses are included as part of the imitation losses during the training loop similar to the other losses as described in section 5.3.
>
> “Past motion dropout”: We keep only the current position of the agent in the past_agent_poses channel. This is a fixed point (u_0, v_0) in the top-down view. The other remaining input channels remain unchanged.
>
> “Figure 6”: Thanks for pointing this out! We have removed this reference from the main text.
>
> “Perturbation about vertical axis”: This is only done during training as a data augmentation mechanism. Clarified in section 6.2.
>
> “Open- and closed-loop”: Consider a sequence of logged data obtained from an expert driving the car. At each timestep in this sequence, we have information about the agent’s past trajectory and we can use this (along with the other features) to create an input for the network. The network then predicts a set of waypoint coordinates which can then be compared to the actual future trajectory of the agent from the log. The L2 distance between the predicted and ground-truth waypoints is plotted in Fig. 5 as the open loop evaluation. Note that at each time step in this evaluation, the agent follows the logged pose exactly and we are just evaluating the network predictions against the ground-truth. In a closed-loop setting, we would like to actually drive the agent using the predicted trajectory and for this, we first convert the predicted trajectory to a set of controls and then use a vehicle simulator to drive the agent forward. This will drive the agent independent of the log from this point forward ultimately generating poses which were never seen in the log. This can thus put the input to the network outside the training distribution thus leading to poor predictions and hence poor driving behavior. However, this is what we care about and hence the evaluations in closed-loop settings are crucial. We have also clarified this in the revised paper.
>
> <response continued below>

---

> > ### Author Response · Authors · 2018-11-10
> > **Author Response (Part 2 of 2)**
> >
> >
> > “Evaluation”: As mentioned in the paper, we evaluate on specific complex situations to illustrate the difference in performance between the different models. This also makes it easy to see how specific non-learned baselines would perform in the said situations.
> >
> > For example, a baseline predicting the route:
> > - [Nudging] Would collide with the parked vehicle in all cases (100% collision) since the route intersects with the parked vehicle, and there is no speed modulation to bring it to a stop.
> > - [Traj Perturb] Would not be able to recover from the trajectory perturbation as the route is no longer under the agent at the starting point and the controller would not be able to execute a point jump (100% stuck).
> > - [Slow Car] Would collide with the slow car since there is no speed modulation information in the route (100% collision).
> >
> > Similarly, a baseline continuing to drive the agent as before:
> > - [Nudging] Would collide with the parked vehicle as it would not slow down at all (100% collision).
> > - [Traj Perturb] Would continue along the perturbed trajectory and go offroad (100% stuck).
> > - [Slow Car] Would collide with the slow car again since it would not slow down at all (100% collision).
> >
> > We are not aware of any other baselines that would help the understanding of these results but are open to adding them if the reviewer finds them useful.
> >
> > “Related Work”: These works relate more to the prediction of other agent’s trajectories to aid decision making for the self-driving car. Our focus in this paper is on the prediction of a drivable trajectory for the self-driving car and then using this trajectory in a closed-loop setting to actually drive the car. However, we agree that these are important references to discuss in the related work to highlight this distinction and we have added them in the revised version.

---

> > > ### Comment · AnonReviewer3 · 2018-11-26
> > > **Reviewer response**
> > >
> > > I would like to thank the authors for their feedback and the various updates to the paper, the text is definitely clearer now.
> > >
> > > When it comes to evaluation, I am still not convinced that the proposed baselines would not be useful. E.g., Figure 5 results could have these extra baselines included and their displacement error could help quantify the proposed method.
> > > Nevertheless, I would like to stay with my recommendation, as I feel that the paper would be interesting to the community, however with certain flaws as pointed out in my review and especially by other reviewers.

---

### Official Review · AnonReviewer2 · 2018-11-05
**Flawed Approach with Poor Results**

**Rating:** 3
**Confidence:** 4

**Review:**

The paper describes a framework for training a self-driving policy by augmenting imitation loss with additional loss terms that penalize undesired behaviors and that encourage progress. The policy takes as input a parsed representation of the scene (rather than raw images) and outputs pose trajectories for a down-stream controller. The method is trained on simulated data that includes perturbations to improve generalizability. The framework is evaluated in simulation through a series of ablations to better understand the contribution of the different loss terms.


STRENGTHS

+ Paper acknowledges the difficulty of end-to-end (pixels-to-torque) learning for autonomous driving and instead reasons over pre-processed inputs in the form of lower-dimensional images (and image sequences) that capture obstacles as bounding boxes and simple lines for routes, grayscale intensities, etc. Similarly, the output is a trajectory that is then fed to a controller responsible for tracking this trajectory.


WEAKNESSES

- The insufficiency of behavioral cloning is not surprising, as noted, given the covariate shift. It would be interesting to consider a  no-regret formulation analogous to Ross et al., 2011, even though it would require interaction with a human.

- The limitation of producing paths in this way is that the network does not explicitly reason over the feasibility of the path, which is important for non-holonomic vehicles. Instead, the network must learn the kinematic and dynamic constraints.

- Perturbations of the simulated trajectories are used to expose the model to collisions and other rare events, but is not clear that simple trajectory perturbations such as those used here provide a sufficient exposure to these rare events.

- The fact that the 2D image that expresses the vehicle's position is absolute limits the environment in which the network is valid. The experiments are conducted on images corresponding to an 80m x 80m environment, which is trivially small.

- The proposed framework is highly specific to self-driving and the extent to which it provides insights for other domains is not clear.

- The ablation experiments are not very compelling. In the case of the nudging experiment, all models result in collisions with M4 being the best model with a 10% collision rate. The trajectory perturbation results are better. In the case of the slowing experiment, M3 is the only version to not result in collision, whereas M4 collides 5% of the time. It isn't clear than which model is preferable since, while M3 never collides in the case of the slowing down experiment, it collides 45% of the time in the nudging experiment, almost as frequently as the M0 baseline.

- The paper claims that the model was run on a real robot, but there is no experimental evaluation of the results, only a reference to videos. The results of these experiments should be quantified and discussed or the reference to running on a real vehicle should be toned down, if not removed.

- Equation 3 requires knowledge of the ground-truth distribution. How is this determined?

---

> ### Author Response · Authors · 2018-11-10
> **Author Response (Part 1 of 2)**
>
> Thanks for reviewing the paper and for your valuable feedback. We have uploaded a revised version which clarifies several technical details. Responses to your concerns follow:
>
> “No-regret formulation”: As suggested, this would require interaction with an expert. In a real world setting, the car would have to start driving, perform out-of-distribution maneuvers, and have the human driver correct it at every step. This is not a safe setup and instead, one would need to rely on offline data or accurate simulation. The novelty of our approach is to demonstrate that we can learn to drive well in complex scenarios beyond simple lane following from offline expert data without requiring a simulator and without doing RL.
>
> “Path feasibility”: We train this network on time sampled expert driving trajectories and hence the network has seen only valid trajectories that satisfy the kinematic and dynamic constraints of the non-holonomic agent. In addition, to generate synthesized trajectories from the perturbed waypoints, we also employ a non-linear optimizer that obeys the same constraints to ensure that the generated synthesized trajectories are feasible to drive as well. Therefore, the network output is constrained to obey these constraints implicitly. Furthermore, we do not use the output trajectory waypoints directly but rather pass them through the same non-linear optimizer to generate driving controls and to smooth out any pixel noise that may push the output trajectory into the infeasible zone. We have not found an instance in practice where the network has produced an infeasible trajectory. The reviewer is correct that the more structure you impose the more you can do with your data, but the comment is somewhat surprising in light of many contemporary published works [1,2]  that try to learn everything end-to-end. For the driving task, the hardest decisions often are mid-level decisions -- where to drive, how to pass / break / accelerate -- we make these via coarse sampled predictions -- the main goal of our net. Controllers are good for refining those.
>
> [1] Codevilla, Felipe, et al. "End-to-end driving via conditional imitation learning." 2018 IEEE International Conference on Robotics and Automation (ICRA). IEEE, 2018.
> [2] Hecker, Simon, Dengxin Dai, and Luc Van Gool. "End-to-end learning of driving models with surround-view cameras and route planners." European Conference on Computer Vision (ECCV). 2018.
>
> “Trajectory perturbations”: We have presented ablation experiments that clearly demonstrate the benefit of introducing these perturbations along with the corresponding loss terms. Specifically, we have shown that these help in (a) handling the covariate shift during closed loop control by teaching the network the idea of course correction, and (b) teaching the network the idea of collision avoidance without having expert demonstrations of the same in the training data. Furthermore, we introduce the general idea of synthesizing perturbations and if required, more complex kinds of perturbations can be introduced to the network in a similar fashion. We show that perturbations help, but expecting them to handle the long tail is too high of a bar that no imitation learning based system would meet.
>
> “Environment extent”: We run the network at 5Hz and since the coordinate system of the top-down view moves with the agent, we found it to be sufficient for control on surface streets. Our predicted trajectory goes to 2s into the future and is sampled every 0.2s. We see 64m ahead of the agent and at 25 mph this gives us a time horizon of 5.72s. We do not see any fundamental flaw in this approach since it is akin to having a self-driving car with sensors that see only up to 64 meters. In fact, most teams on the DARPA Urban Challenge employed the Velodyne HDL-64E as their primary sensor which had an effective vehicle detection range of only 60m [3]. The only practical limits we have found from this range are in cases like T-junctions where we have limited visibility on the sides. However, this is not a limitation of the core concepts presented in the paper -- we can easily increase the extent of the input image at the cost of additional computation.
>
> [3] Montemerlo, Michael, et al. "Junior: The stanford entry in the urban challenge." Journal of field Robotics 25.9 (2008): 569-597.
>
> “Self-driving”: Our work is about getting imitation learning to the level where it has a shot at driving a real vehicle; although the same insights may apply to other domains, these domains might have different constraints and opportunities, so we do not want to claim contributions there. We have revised the paper to reflect this. Furthermore, we believe that the self-driving domain is an area of broad interest for the machine learning research community as is evident from the surge in papers on using machine learning techniques for planning and prediction in this domain.
>
> <response continued below>

---

> > ### Author Response · Authors · 2018-11-10
> > **Author Response (Part 2 of 2)**
> >
> >
> > “Ablation experiments”: As noted in the paper, we found that simple driving scenarios are handled well by our model and the ablation experiments thus focus on complex driving behaviors, which illustrate the gains from the techniques introduced in the paper. Specifically for the nudging experiment, we generate several variations by changing the starting speed of the agent relative to the parked car. This creates situations of increasing difficulty, where the agent approaches the parked car at very high relative speed and thus does not have enough time to nudge around the car given the dynamic constraints. A 10% collision rate in this case is thus not a measure of the absolute performance of the model since we don’t have a perfect driver which could have performed well at all the scenarios here. The focus of our experiments is on the relative improvement in performance across models. M3 and M4 only differ in the use of either a weighted or a dropout strategy for combining imitation vs environment losses. In the case of the slowing down experiment, M4 collides with the curb (not the vehicle) in one scenario in trying to pass the slow vehicle (again for the variation at the extreme initialization of approaching at the highest relative speed) but we find that it does much better in all other scenarios and is thus our proposed model. The fact that M3 collides 45% of the time in the nudging experiments points to its inability to deal with real collisions.
> >
> > “Real driving experiments”: Being able to drive a real vehicle with this approach is the most exciting aspect of this work -- no other imitation learning system has ever accomplished this. Most recent works demonstrate the performance of their setup either in an open-loop setting or in a closed-loop simulation setup like CARLA which does not suffer from challenges like controller errors, actuator delays and perception errors. Our video results from the real-drive illustrate not only the smoothness of the network’s driving ability, but also its ability to deal with stop-signs and turns and to drive for long durations in full closed-loop control without deviating from the trajectory (as would be the case if one were to use pure behavior cloning). Performing quantitative evaluations in the real world safely, and developing the right metrics to compare against other planners including classical planners remains future work.
> >
> > “Equation 3”: The ground truth distribution P_k^{gt} is simply a dirac delta function with a value of 1 at the spatial coordinate of the target waypoint and zero everywhere else. The loss L_p = H(P_k, P_k^{gt}) thus measures the cross-entropy between the two probability distributions where P_k is the predicted distribution. We have clarified this in the revised version.

---

### Meta-Review · Area_Chair1 · 2018-12-13

**Confidence:** 5
**Recommendation:** Reject

**Metareview:**

The authors present a method for training a policy for a self-driving car. The inputs to the policy are map-based perceptual features and the outputs are waypoints on a trajectory, and the method is an augmented imitation learning framework that uses perturbations and additional losses to make the policy more robust and effective in rare events. The paper is clear and well-written and the authors do demonstrate that it can be used to control a real vehicle. However, the reviewers all had concerns about the oracle feature representation which is the input and also concerns about the lack of baselines such as optimization based methods. They also felt that the approach was limited to self-driving cars and thus would have limited interest for the community.